# CrossLoco: Human Motion Driven Control of Legged Robots via Guided Unsupervised Reinforcement Learning

**Tianyu Li, Hyunyoung Jung, Matthew Gombolay, Yong Kwon Cho, Sehoon Ha**
Georgia Institute of Technology
Atlanta, GA 30332, USA
`{tli471,hjung331,sehoonha}@gatech.edu`
`matthew.gombolay@cc.gatech.edu`
`yong.cho@ce.gatech.edu`

## Abstract

Human motion driven control (HMDC) is an effective approach for generating natural and compelling robot motions while preserving high-level semantics. However, establishing the correspondence between humans and robots with different body structures is not straightforward due to the mismatches in kinematics and dynamics properties, which causes intrinsic ambiguity to the problem. Many previous algorithms approach this motion retargeting problem with unsupervised learning, which requires the prerequisite skill sets. However, it will be extremely costly to learn all the skills without understanding the given human motions, particularly for high-dimensional robots. In this work, we introduce *CrossLoco*, a guided unsupervised reinforcement learning framework that simultaneously learns robot skills and their correspondence to human motions. Our key innovation is to introduce a cycle-consistency-based reward term designed to maximize the mutual information between human motions and robot states. We demonstrate that the proposed framework can generate compelling robot motions by translating diverse human motions, such as running, hopping, and dancing. We quantitatively compare our *CrossLoco* against the manually engineered and unsupervised baseline algorithms along with the ablated versions of our framework and demonstrate that our method translates human motions with better accuracy, diversity, and user preference. We also showcase its utility in other applications, such as synthesizing robot movements from language input and enabling interactive robot control.

## 1 Introduction

The concept of teleoperating robots through human movements, known as Human Motion Driven Control (HMDC), has been illustrated in various forms of media, including animations, movies, and science fiction, such as Madö King Granzört (Iuchi, 1989), Pacific Rim (del Toro, 2013), and Ready Player One (Spielberg, 2018). In these media, HMDC technology allows operators to intuitively control robots using their body movements. Compared to fully autonomous control, this teleoperation offers the essential dexterity and decision-making capabilities required for tasks demanding precise motor skills and situational awareness. Consequently, this property makes HMDC promising for various applications, including entertainment, medical surgery, and space exploration.

The key challenge of HMDC is how to establish the correspondence between robot states and human motions, which can also be referred to as motion retargeting. For certain types of robots, such as humanoids or manipulators, this correspondence might be simple enough to be approached by assuming the mapping of end-effectors in Cartesian space and solving the formulated inverse kinematics problem (Gleicher, 1998; Tak & Ko, 2005). However, when we consider robots with significantly different morphological structures, such as quadrupeds, hexapods, or quadrupeds with mounted arms, the correspondence becomes nontrivial due to the intrinsic ambiguity of the problem. Therefore, researchers often have approached this motion retargeting problem by applying supervised learning techniques to the paired datasets (Sermanet et al., 2018; Delhaisse et al., 2017; Rhodin et al., 2014). Nonetheless, creating paired datasets can be a challenging and labor-intensive

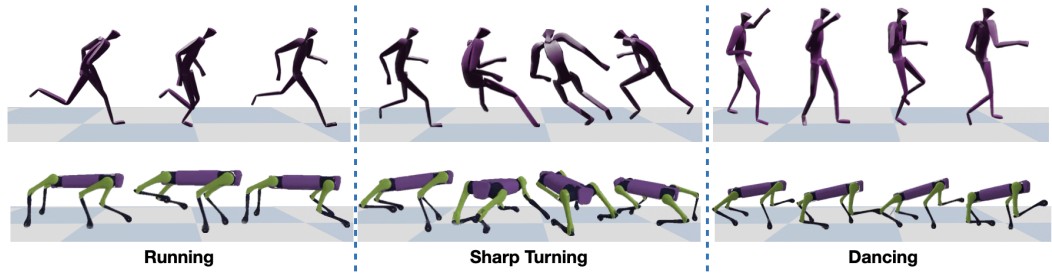

Figure 1: We introduce *CrossLoco*, a guided unsupervised reinforcement learning framework for translating human motion to robot control.

task that requires significant engineering expertise. To address this issue, some researchers have proposed using unsupervised learning techniques to learn the correlation from unpaired human and robot motion datasets (Li et al., 2023b; Choi et al., 2020; Smith et al., 2019). In this case, the robot dataset serves as prior knowledge indicating the motion pattern of the robot. However, obtaining motion datasets can be expensive because we do not know the required skills for the given human motion. In addition, control itself is challenging due to the complexity of the quadrupedal robot and its underactuated dynamics. This leads to our research question: can we learn cross-morphology HMDC without prior knowledge of the robot?

The research question presents three primary challenges. Firstly, the significant difference in kinematics and dynamics between the human and the target robot makes it difficult to establish correspondence. Secondly, we cannot build a predefined motion database for the robot due to the complexity of the problem. Finally, the problem itself is ambiguous. For instance, there exist many different quadrupedal gaits that can capture the essence of human walking. To address these challenges, we drew inspiration from the recent unsupervised skill discovery techniques, such as Eysenbach et al. (2018) and Peng et al. (2022), and aim to simultaneously learn robot skills and robot-human motion correspondence by maximizing the mutual information between human and robot motions.

In this work, we introduce *CrossLoco*, a guided unsupervised reinforcement learning framework that enables simultaneous learning of human-robot motion correspondence and robot motion control (Figure 1). Our key approach is to introduce a cycle-consistency-based correspondence reward term that maximizes the mutual information between human motions and the synthesized robot movements. We implement this cycle consistency term by training both robot-to-human and human-to-robot reconstruction networks. Our formulation also includes regularization terms and a root-tracking reward to guide correspondence learning. Simultaneously, we train a robot control policy that takes human motions and sensory information as input and generates robot actions for interacting with the environment.

We demonstrate that *CrossLoco* can translate a large set of human motions for robots, including walking, running, and dancing. Even for locomotion, the robot exhibits two distinct strategies, trotting and galloping, inspired by human walking motions with different styles. We quantitatively compare our method against the baseline, DeepMimic (Peng et al., 2018a), along with the ablated versions of our *CrossLoco* framework and show that our method can achieve better quantitative results in terms of accuracy, diversity, and user preference. We further showcase the potential applications of our framework: **language2text** motion synthesis and **interactive motion control**.

## 2 RELATED WORKS

**Learning Locomotion Skills**. There are various methods for robots to learn locomotion skills. One approach involves maximizing a reward function designed by experts using reinforcement learning, as demonstrated in several studies such as Tan et al. (2018); Haarnoja et al. (2018); Xie et al. (2018); Li et al. (2019); Rudin et al. (2022). Another method is motion imitation, where the control policy is trained with an imitation reward, as shown in Peng et al. (2018a;b); Li et al. (2023a); Bergamin et al. (2019); Won & Lee (2019); Ling et al. (2020); Peng et al. (2020). This reward is calculated based on the distance between the robot's current pose and a reference pose from the demonstration

trajectory. The closer the distance, the larger the reward. Generative adversarial imitation learning (GAIL) (Ho & Ermon, 2016) is another approach that trains the policy to deceive a discriminator that distinguishes real and fake demonstration data. Finally, without the need for engineering reward functions or demonstration data, some studies such as Eysenbach et al. (2018); Sharma et al. (2019) focus on unsupervised skill discovery from interaction data through information-theoretic methods.

**Motion Retargeting**. Transferring motions between different morphologies has been an important topic in both robotics and computer graphics communities to produce natural motions for various robots and characters. Researchers have investigated various approaches, such as designing manual correspondences (Gleicher, 1998; Tak & Ko, 2005; Grandia et al., 2023), learning from paired datasets (Sermanet et al., 2018; Delhaisse et al., 2017; Jang et al., 2018), or developing modular/hierarchical policies (Won & Lee, 2019; Hejna et al., 2020; Sharma et al., 2019). More recent works (Zhang et al., 2020; Aberman et al., 2020; Villegas et al., 2018; Li et al., 2023b; Smith et al., 2019; Kim et al., 2020; Shankar et al., 2022) aim to learn the state and action correspondence from unpaired datasets via unsupervised learning. However, these methods often require a pre-collected dataset of both domains, which is not available for robots in our problem.

**Cycle-Consistency**. Our work is inspired by previous research on cycle-consistency (Zhou et al., 2016; Zhu et al., 2017; Liu et al., 2017; Rao et al., 2020; Bousmalis et al., 2018). For instance, CycleGAN (Zhu et al., 2017) combines cycle-consistency loss with Generative Adversarial Networks (Goodfellow et al., 2014) for unpaired image-to-image translation. By adding domain knowledge, CycleGAN can be extended to video retargeting (Bansal et al., 2018) and domain adaptation (Hoffman et al., 2018). In robotics, a similar approach has been investigated for sim-to-real transfer (Stein & Roy, 2018; James et al., 2019). Besides alignment in image space, a few researchers (Zhang et al., 2020; Shankar et al., 2022) adopt cycle-consistency to align agents in different dynamics and structures, while the others (Aberman et al., 2020; Villegas et al., 2018) apply cycle-consistency for motion retargeting between similar human-like robots or characters. Inspired by these works, we aim to co-train a control policy for the diverse motor skills of a quadrupedal robot while establishing cycle consistency between the robot and human motions.

## 3 PRELIMINARIES

**Skill-Conditioned Reinforcement Learning.** We formulate our framework as a skill-conditioned reinforcement learning problem, where an agent interacts with an environment to maximize an objective function by following a policy $\pi$. At the beginning of each learning episode, a condition term is sampled from the dataset $\mathbf{z} \sim p(\mathbf{z})$. At each time step, the agent observes the state of the system $\mathbf{s}_t$, then takes an action sampled from the policy $\mathbf{a}_t \sim \pi(\mathbf{a}_t|\mathbf{s}_t, \mathbf{z})$ to interacts with the environment. After executing the actions, the environment takes the agent to a new state sampled from the dynamics transition probability $\mathbf{s}_{t+1} \sim p(\mathbf{s}_{t+1}|\mathbf{s}_t, \mathbf{a}_t)$. A scalar reward can be measured using a reward function $r_t = r(\mathbf{s}_t, \mathbf{a}_t, \mathbf{s}_{t+1}, \mathbf{z})$. The agent's objective is to learn a policy that maximizes its expected cumulative reward $J(\pi)$,

$$J(\pi) = \mathbb{E}_{\mathbf{z} \sim p(\mathbf{z}), \boldsymbol{\tau} \sim p(\boldsymbol{\tau}|\pi, \mathbf{z})} [\sum_{t=0}^{T-1} \gamma^t r_t]. \tag{1}$$

Here, $\boldsymbol{\tau}$ is a state and action trajectory with the length $T$, where its distribution can be computed as $p(\boldsymbol{\tau}|\pi, \mathbf{z}) = p(\mathbf{s}_0) \prod_{t=0}^{T-1} p(\mathbf{s}_{t+1}|\mathbf{s}_t, \mathbf{a}_t)\pi(\mathbf{a}_t|\mathbf{s}_t, \mathbf{z})$ is the likelihood of the trajectory under policy $\pi$. The initial state $\mathbf{s}_0$ is sampled from the distribution $p(\mathbf{s}_0)$ and $\gamma \in [0, 1)$ is a discount factor.

**Skill Discovery By Maximizing Mutual Information.** Eysenbach et al. (2018) and Peng et al. (2022) formulate the skill discovery problem as an unsupervised reinforcement learning problem, where the objective is to maximize the mutual information between the robot state and a latent vector sampled from a distribution $\mathbf{z} \sim p(\mathbf{z})$: $I(\mathbf{S}; \mathbf{Z}) = H(\mathbf{S}) - H(\mathbf{S}, \mathbf{Z})$. This equation can be interpreted as the policy $\pi$ is to learn to produce diverse behaviors while each latent vector $\mathbf{z}$ should correspond to distinct robot states.

However, this equation is intractable in most scenarios where the state marginal distribution is unknown, and two tricks are commonly implemented to tackle this. The first trick is to take advantage of the symmetry of mutual information:

$$I(\mathbf{S}; \mathbf{Z}) = I(\mathbf{Z}; \mathbf{S}) = H(\mathbf{Z}) - H(\mathbf{Z}|\mathbf{S}). \tag{2}$$

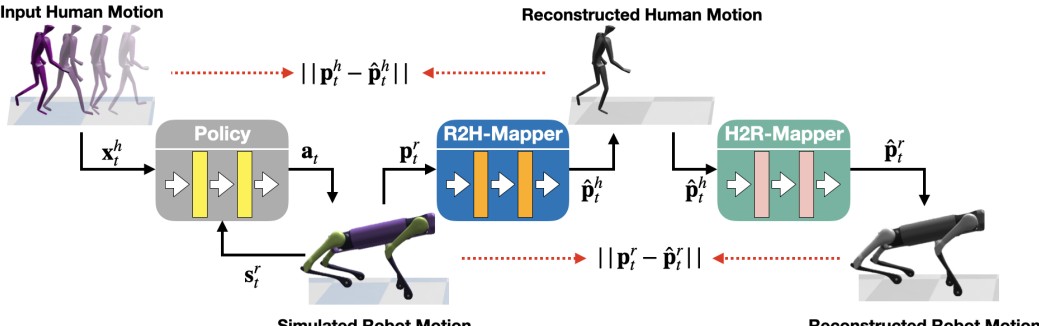

Figure 2: Method Overview. A robot control policy utilizes human motions and sensory information to generate robot actions for interacting with the environment. The Robot2Human-Mapper is then used to reconstruct the input human motion pose from the robot's pose. Lastly, the Human2Robot-Mapper is used to reconstruct the robot pose from the reconstructed human pose. Both mappers are trained via supervised learning using the difference between the real and reconstructed poses. These differences are also utilized for constructing a correspondence reward function, which is used to update the policy (see Equation 8).

This trick removes the need for measuring the marginal entropy of robot state $H(\mathbf{S})$ by instead measuring the entropy of the latent vector $H(\mathbf{Z})$ which remains constant in fixed skill prior $p(\mathbf{z})$. The second trick is to use a variational lower bound as proposed by Eysenbach et al. (2018) and Gregor et al. (2016) to approximate the mutual information as follows:

$$I(\mathbf{Z}; \mathbf{S}) = H(\mathbf{Z}) - H(\mathbf{Z}|\mathbf{S}) \geq \max_q H(\mathbf{z}) + \mathbb{E}_{\mathbf{z} \sim p(\mathbf{z}), \mathbf{s} \sim p(\mathbf{s}|\pi_z)}[\log(q(\mathbf{z}|\mathbf{s})], \tag{3}$$

where $q(\mathbf{z}|\mathbf{s})$ is a variational approximation of the conditional distribution $p(\mathbf{z}|\mathbf{s})$ and the lower bound is tight if $q = p$. This skill discovery objective encourages a policy to produce distinct behaviors for different skill vectors $\mathbf{z}$ by designing a reward based on the measurement of $q(\mathbf{z}|\mathbf{s})$.

# 4 CROSSLOCO

*CrossLoco* is a guided unsupervised reinforcement learning framework designed to learn robot locomotion control policy driven by human motion. The framework establishes a correspondence between human and robot motions, enabling the robot to acquire locomotion skills from human motions. The method overview is shown in Fig 2. In this section, we first introduce how we formulate the problem. Then, we present our cycle-consistency-based method for learning locomotion and human-robot correspondence. Lastly, we provide additional implementation details.

## 4.1 PROBLEM FORMULATION

Our goal is to train a robot control policy, denoted as $\pi$, that can produce various robot motions based on different human motion inputs. Therefore, we can view our problem as a Markov Decision Process conditioned on the given human motion. Let us define $\mathbf{p}_t^h$ and $\mathbf{p}_t^r$ as the human and robot kinematic poses. Then the human motion is defined as a sequence of pose vectors: $\mathbf{m}^h = [\mathbf{p}_0^h, \mathbf{p}_1^h, \cdots, \mathbf{p}_{T-1}^h]$. The robot state $\mathbf{s}_t^r$ represents both the kinematic and dynamic status of the robot, hence we can view $\mathbf{s}_t^r$ as the superset of the pose $\mathbf{p}_t^r$. The robot action $\mathbf{a}_t$ corresponds to motor commands, such as target joint angles. At each time step, the policy takes the robot state vector $\mathbf{s}_t^r$ and the augmented human motion feature $\mathbf{x}_t^h = x(\mathbf{p}_t^h)$ as input to generate an action $\mathbf{a}_t \sim \pi(\mathbf{a}_t|\mathbf{s}_t^r, \mathbf{x}_t^h)$. Then our goal is to maximize the given reward function $r$:

$$J(\pi) = \mathbb{E}_{\mathbf{m}^h \sim p(\mathbf{m}^h), \boldsymbol{\tau}^r \sim p(\boldsymbol{\tau}^r|\pi, \mathbf{x}_t^h)} \Big[ \sum_{t=0}^{T-1} \gamma^t r(\mathbf{s}_t^r, \mathbf{x}_t^h, \mathbf{a}_t) \Big], \tag{4}$$

where the robot trajectory is defined as a sequence of the robot states $\boldsymbol{\tau}^r = [\mathbf{s}_0^r, \mathbf{s}_1^r, \cdots, \mathbf{s}_{T-1}^r]$.

This formulation leads to the question of designing an effective reward function $r$ that builds the relationship between the human and robot poses, $\mathbf{p}_t^h$ and $\mathbf{p}_t^r$. In addition, the reward function should

---

**Algorithm 1** CrossLoco pseudocode

---

**Require:** Human Dataset $\mathcal{M}$.
1: Initialize: Policy $\pi$, Value function $V$, *R2H-Mapper* $q^{r2h}$, *H2R-Mapper* $q^{h2r}$, Data Buffer $\mathcal{D}$.
2: **repeat**
3:     **for** trajectory i = 1, ..., m **do**
4:         $\mathcal{D} \leftarrow \{(x_t^h, s_t^r, a_t, r_t)_{t=0}^{T-1}\}$ collect trajectory by rolling out $\pi$.
5:     update $\pi$ and $V$ using PPO with data from $\mathcal{D}$.
6:     update $q^{r2h}$ and $q^{h2r}$ using data from $\mathcal{D}$ by minimizing $L^{r2r}$ (Equation 7).
7:     Reinitialize Data Buffer $\mathcal{D}$.
8: **until** Done

---

include some regularization terms, such as minimizing the energy, avoiding self-collisions, or preserving the predefined semantic features. We will discuss the design of our reward function in the following section.

## 4.2 MEASURING CORRESPONDENCE VIA CYCLE-CONSISTENCY

To develop a reward function that represents the correspondence between human and robot motions, we borrow the information-theoretic approach mentioned in the previous section. We formulate the correspondence reward term such that it maximizes the mutual information between human and robot pose, given by $I(\mathbf{p}_t^r, \mathbf{p}_t^h | \pi)$. From Equation 3, this formulation can be approximated by:

$$I(\mathbf{p}_t^r, \mathbf{p}_t^h | \pi) \geq H(\mathbf{p}_t^h) + \mathbb{E}_{\mathbf{m}^h \sim p(\mathbf{m}^h), \mathbf{p}_t^r \sim p(\mathbf{p}_t^r | \mathbf{p}_t^h, \pi)}[\log(q^{r2h}(\mathbf{p}_t^h | \mathbf{p}_t^r))]], \tag{5}$$

where we assume that human motion prior is from the fixed dataset and refer to $q^{r2h}$ as Robot-to-Human Mapper (*R2H-Mapper*). Because the first term $H(\mathbf{p}_t^h)$ is constant, we can find the optimal policy by maximizing the second term, $\log[q^{r2h}(\mathbf{p}_t^h | \mathbf{p}_t^r)]$. A higher value represents that *R2H-Mapper* is more certain about the human pose given the robot pose, hence indicating that the human pose is distinctive given the robot pose.

We model the *R2H-Mapper* as a Gaussian distribution with fixed covariance $q^{r2h}(\mathbf{p}_t^h | \mathbf{p}_t^r) = N(\mu^{r2h}(\mathbf{p}_t^r), \sigma)$ where $\mu^{r2h}(\mathbf{p}_t^r)$ is the mean of the distribution while $\sigma$ is the constant covariance matrix. The *R2H-Mapper* can be trained by minimizing a loss function $L^{r2h}$:

$$\underset{q^{r2h}}{\arg\min} \; L^{r2h} = \mathbb{E}_{\mathbf{p}_t^h \sim p(\mathbf{p}_t^h), \mathbf{p}_t^r \sim d^\pi(\mathbf{p}_t^r | \mathbf{p}_t^h)}[||\mathbf{p}_t^h - \mu^{r2h}(\mathbf{p}_t^r)||_2^2], \tag{6}$$

where $d^\pi(\mathbf{p}_t^r | \mathbf{p}_t^h)$ is the likelihood of observing robot pose $\mathbf{p}_t^r$, by executing policy $\pi$ given the human pose $\mathbf{p}_t^h$. Similarly, we can design our correspondence reward to minimize the given term $||\mathbf{p}_t^h - \mu^{r2h}(\mathbf{p}_t^r)||_2^2$.

However, *R2H-Mapper* does not prevent multiple robot poses $\mathbf{p}^r$ from being mapped to the same human pose $\mathbf{p}^h$ because it only considers one-directional mapping, which may cause degenerated motions. To address this issue, we add a Human-to-Robot Mapper (*H2R-Mapper*), denoted as $q^{h2r}(\mathbf{p}_t^r | \mathbf{p}_t^h)$, which is used for mapping the human pose back to the robot pose. We use a cycle-consistency formulation of Zhu et al. (2017), where we first map the robot pose to the human pose, followed by mapping the generated human pose back to the robot pose. This results in an objective loss function $L^{r2r}$ for *H2R-Mapper* and *R2H-Mapper* as:

$$\underset{q^{r2h}, q^{h2r}}{\arg\min} \; L^{r2r} = L^{r2h} + \mathbb{E}_{\mathbf{p}_t^h \sim p(\mathbf{p}_t^h), \mathbf{p}_t^r \sim d^\pi(\mathbf{p}_t^r | \mathbf{p}_t^h)}[||\mathbf{p}_t^r - \mu^{h2r}(\mu^{r2h}(\mathbf{p}_t^r))||_2^2]. \tag{7}$$

Finally, from our mutual information maximization and cycle-consistency loss minimization, we formulate the correspondence reward as follows:

$$r_t^{cpd} = \exp(-||\mathbf{p}_t^h - \mu^{r2h}(\mathbf{p}_t^r)||_2^2 - ||\mathbf{p}_t^r - \mu^{h2r}(\mu^{r2h}(\mathbf{p}_t^r))||_2^2). \tag{8}$$

## 4.3 IMPLEMENTATION DETAILS

During the training process, the policy $\pi$, as well as the H2R-Mapper and R2H-Mapper, are updated iteratively. The policy is trained using Proximal Policy Optimization (PPO) Schulman et al. (2017).

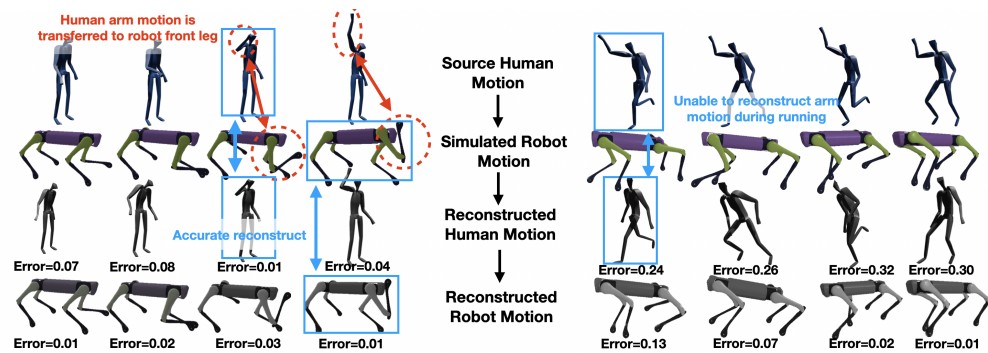

Figure 3: Visualization of reconstructed human and robot pose with reconstruction error. We show that the mappers can accurately reconstruct human and robot poses when the robot actively follows human movement. When the human is relatively stationary, human arm movements can be mapped onto the robot's front legs. In scenarios like human running, the framework focuses primarily on leg motion.

Meanwhile, the H2R-Mapper and R2H-Mapper are trained using supervised learning. The learning framework is summarized in Algorithm 1.

**Model Representation.** Human pose ($\mathbf{p}^h \in R^{23}$) and robot pose ($\mathbf{p}^r \in R^{17}$) consist of local information, including root height, root orientation, and joint pose. The robot state ($\mathbf{s}^r_t \in R^{47}$) contains all the information in $\mathbf{p}^r$, as well as root and joint velocity, and previous action. The human feature vector ($\mathbf{x}^h_t \in R^{188}$) includes human pose, root velocity, and joint velocity information at the future 1, 2, 10, and 30 frames.

**Complete Reward Function.** In addition to the correspondence reward mentioned earlier, our reward function includes several terms to regulate the training and preserve high-level semantics. A root tracking reward, denoted as $r^{root}_t = exp(-||\mathbf{s}^{root}_t - \bar{\mathbf{s}}^{root}_t||)$, is designed to preserve high-level movements by minimizing the deviation between the normalized base trajectories of the human ($\bar{\mathbf{s}}^{root}_t$) and the robot ($\mathbf{s}^{root}_t$). In this context, both trajectories include the root position and height, which are normalized by their respective leg lengths. Without this term, the resulting correspondence can be arbitrary: e.g., a human forward walking motion can be mapped into a robot's lateral movements. To prevent unrealistic movements, a torque penalty, $r^{tor}_t = -||\mathbf{a}_t||$, and joint limits penalty, $r^{lim}_t = -\mathbf{1}_{\mathbf{p}^r_t > \mathbf{p}^{lim}}$, are borrowed from Rudin et al. (2022). Here, $\mathbf{p}^{lim}$ is the pose limit of the robot. The overall reward is calculated as the weighted sum of all these terms: $r_t = w^{cpd}r^{cpd}_t + w^{root}r^{root}_t + w^{tor}r^{tor}_t + w^{lim}r^{lim}_t$. To optimize the weights, increasing $w^{cpd}$ can improve the correspondence between robot and human motion, but it may negatively affect root tracking performance or increase energy consumption. Increasing $w^{root}$ puts more emphasis on root tracking. Small values for $w^{tor}$ and $w^{lim}$ can result in unnatural motions, while excessively large values can lead to overly conservative motions. In our setting, we use $[w^{cpd}, w^{root}, w^{tor}, w^{lim}] = [1.0, 1.0, 0.0001, 5.0]$.

**Network Structure.** The policy, critic, H2R-Mapper, and R2H-Mapper are modeled by a fully-connected network consisting of three hidden layers with 512 nodes each. ELU is used as the activation function for the policy and critic, while ReLU is used for the mapper networks.

## 5 EXPERIMENTS

We conduct a series of experiments to investigate three key aspects of the proposed work: firstly, the feasibility of acquiring a human-motion-driven robot controller, which is referred to as the '**human2robot**' controller; secondly, the comparative performance against alternative baseline approaches; and lastly, the influence of the correspondence reward on the training process.

We evaluate the effectiveness of our approach by transferring a set of human motions to Aliengo quadrupedal robot (unitree, 2023) with 12 joints, which has a significantly different morphology compared to humankind. We take human motion from LaFAN1 dataset (Harvey et al., 2020). Our human dataset consists of 50 human motion trajectories with eight seconds of the average clip length. The dataset contains various types of human movements, including walking, running, hopping, and

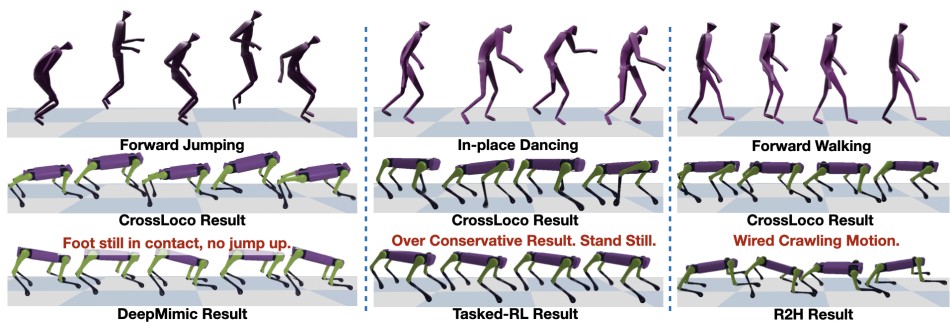

Figure 4: Comparison of generated motions. Our method, *CrossLoco*, synthesizes more compelling motions compared to the baselines by preserving both high-level semantics and fine motion details.

dancing. All experiment environments are conducted using Isaac Gym (Makoviychuk et al., 2021), a high-performance GPU-based physics simulator. During training, 1024 environments are simulated in parallel on a single NVIDIA GeForce RTX 3080 Ti GPU for a period of about 3 days.

## 5.1 Main Results and Analysis

We present the results of our human motion driven control experiments in Figure 4. Our method successfully learns a *human2robot* controller that can transfer various human motions to a robot with a different morphology. We observe agile motions from the robot, such as running and sharp turning when the human performs fast locomotion. The results also demonstrate that the robot can creatively follow human dancing motions, which is hard to manually design. This showcases the capability of our method to establish automatic correspondence between humans and robots while learning diverse robot skills. All the motions can be best seen in the supplementary video.

In Figure: 3, we demonstrate the effectiveness of our proposed correspondence reward in conjunction with the R2H and H2R mappers, which enable the robot to mimic human movements accurately. It's important to note that such synchronization is unattainable without the correspondence reward. Without it, the robot would remain stationary or merely track the human's root position. Our experiments have revealed noteworthy findings. For instance, even when humans walk at a similar speed but with varying styles—such as different step frequencies—the robot adapts its movement to match the human's frequency. Another intriguing observation is the relationship between the correspondence and root tracking. When a human is stationary or moves minimally, the robot responds effectively to the human's arm movements. However, during rapid movement, the robot prioritizes leg motion. We hypothesize that during high-speed locomotion, the root tracking reward becomes more dominant, leading to a balancing act in the CrossLoco framework between root tracking and achieving high-quality correspondence.

In some scenarios, the robot is not able to perfectly mimic the given human motion. For instance, when a human moves backward and makes a sharp 180-degree turn, the robot cannot follow the desired orientation. Additionally, *CrossLoco* may struggle to transfer large side stepping. There are two potential reasons for these imperfections. First, the neural network may be incapable of capturing all motions. Second, the robot's morphology may prohibit it from performing certain motions that are easy for humans, such as swift turning. During training, we observed that the diversity of human motion is critical to the training outcome. This is because the learned mappers can overfit to specific scenarios. For instance, if the dataset contains only hand-waving motions, the robot might slightly vibrate the root to maximize the correspondence reward hence lead to undesirable motion transfer results.

## 5.2 Baseline Comparison

We further quantitatively compare our method against the following baseline methods:

- **Engineered Motion Retarget + DeepMimic (DeepMimic)**: This baseline contains two stages. Firstly, an expert manually designs a motion retargeting function to translate human motions to robot motion referenced trajectories. Then, the robot is trained to track these

|  | ACR ↑ | DIV ↑ | RTR ↑ | CR ↑ | PR ↑ |
|---|---|---|---|---|---|
| **CrossLoco (Ours)** | **0.785** | **2.853** | **0.743** | 65.5% | **43%** |
| DeepMimic | 0.558 | 2.231 | 0.579 | **67.5%** | 16% |
| Task-Only | 0.556 | 2.494 | 0.740 | 30.0% | 14% |
| R2H-Only | 0.683 | 2.779 | 0.729 | 42.5% | 27% |

Table 1: Quantitative Results. Our *CrossLoco* outperforms all the criteria except for being the second-best at correctness with a small margin.

reference trajectories using DeepMimic (Peng et al., 2018a). It is important to note that the result of this baseline heavily relies on the quality of the retargeted motions, which requires a significant amount of effort from the expert. In our case, we have designed the retargeted motions by matching the human foot and robot foot with a fixed tripod gait.

- **Task-Only**: This baseline is designed to investigate the impact of the proposed correspondence reward on training outcomes. As such, we compare this approach to CrossLoco, where the weight of correspondence is set to zero ($w^x = 0$). Therefore, this policy is trained solely on a root tracking reward and other regularization rewards.

- **R2H-Only**: This baseline is designed by removing the robot pose cycle-consistency part from CrossLoco and only keeping the human pose's consistency. As there is no robot pose consistency, in this baseline, the correspondence reward is defined as $r_t^{cpd,r2h} = \exp(-||\mathbf{p}_t^h - \mu^{r2h}(\mathbf{p}_t^r)||_2^2)$.

Since we assume we have no robot motion dataset, hence, we don't include any GAN-based baseline methods, such as Adversarial Correspondence Embedding (Li et al., 2023b).

Our objective is to quantitatively assess the effectiveness of establishing correspondence between human and robot motion, as well as the diversity of the robot motion of these methods. In order to achieve this, we utilize the following metrics:

- **Averaged Correspondence Reward (ACR)**: This term measures the correspondence between human and robot motions. A higher ACR indicates better correspondence. Even though no correspondence reward is used in each baseline training procedure, we co-train *R2H-Mapper* and *H2R-Mapper* to measure the correspondence reward.

- **Diversity (DIV)**: This term has been used for measuring motion diversity in many SOTA works (Shafir et al., 2023; Guo et al., 2023). A higher DIV indicates robots can acquire more skills. From a set of all generated motions from different source human motions, two subsets of the same size $S_d$ are randomly picked. The diversity of this set of motion is defined as: $DIV = \frac{1}{S_d}\Sigma_{i=1}^{S_d}||\Psi(\mathbf{s}_i^r) - \Psi^r(\mathbf{s'}_i^r)||$. $\Psi(\mathbf{s}_i^r)$ and $\Psi(\mathbf{s'}_i^r)$ are features extracted from robot state. Here, we pick robot root velocity and joint pose as the feature.

- **Averaged Root Tracking Reward (RTR)**: This term measures if the learned policy can track the desired root trajectory. The root tracking reward is defined in Section 4.3.

In addition to these metrics, we conducted a user study with 15 subjects to evaluate the performance from a subjective perspective.

- **Correct Rate (CR):** We first investigate whether users can identify a correct match between the given human and synthesized robot motions. A user is tasked to find a matched pair from one human animation and four retargeted robot motions. One robot motion is retargeted from the given human motion, while the other three are generated from different inputs. We examined the combination of four human motions and four methods, and then measured the percentage of correct matches. Ideally, a good transfer should accurately capture the style of the human motion, resulting in easy matching for the user.

- **Preference (PR):** In the second part, we provided users with robot motions generated with different approaches. We asked the users to select the motions that they believed represented a good transfer. We then measured the ratio at which each method was chosen.

Our results are summarized in Table 1. The quantitative analysis indicates that *CrossLoco* outperforms the baseline methods in all metrics suggesting that it can effectively learn a controller that can translate different human motions to diverse robot motions while tracking the desired root trajectory.

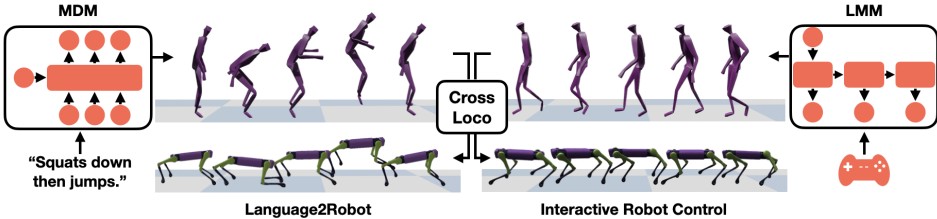

Figure 5: Illustration of two applications: **Language2Robot** and **Interactive Robot Control**.

Although DeepMimic is designed as a one-to-one mapping between human and robot poses, it achieved a lower correspondence reward than *CrossLoco* (0.558 vs 0.785). This could be attributed to the fact that an engineering mapping function may not be physically feasible for all input human motions, and the learning process may sometimes sacrifice the desired posing tracking for a higher desired root tracking reward. Moreover, since the engineered robot desired motion can sometimes be physically infeasible, the root tracking reward of DeepMimic is also lower than that of CrossLoco (0.579 vs 0.743). Based on the results of the user study, it was found that DeepMimic's motion (with a 16% PR score) was not preferred by users. However, it achieved the highest CR score (67.5%). This can be because users can match human and robot motions based on the most obvious frames, even with poor overall motion quality.

As for Task-only, since it is trained only with root tracking and regularization rewards, it produces conservative motions by ignoring human leg motions in some cases, such as different in-place dancing motions. All human in-place dancing motions are mapped to the robot standing with slight root movements by Task-only baseline. However, the correspondence reward in *CrossLoco* triggers robots to learn diverse skills that correspond to different human motions, as evidenced by its superior performance in terms of correspondence reward, diversity term, and user study results compared to Task-only. We also obverse that for *CrossLoco* achieves slightly higher root tracking reward, indicating *CrossLoco*'s great capability.

The results show that CrossLoco achieves a higher correspondence reward and more diverse motion compared to R2H-Only. Additionally, users found the results of CrossLoco to be more distinguishable. This could be attributed to the effective regularization provided by the cycling of human back to robot, resulting in more distinguishable outcomes.

### 5.3 APPLICATIONS

Numerous studies have delved into human motion synthesis using a variety of input sources, such as text (Bahl et al., 2022), music Tseng et al. (2023), and user inputs (Holden et al., 2020). Our learned human2robot controller can be seamlessly integrated with these modules, utilizing human motion as the interface for new applications. In this context, we have implemented two examples: **Language2Robot** and **Interactive Robot Control**. An illustration of these examples is presented in Figure 5. **Language2Robot** merges the established text-to-human-motion module (MDM (Bahl et al., 2022)) with CrossLoco, enabling the generation of robot motion based on verbal instructions. **Interactive Robot Control** combines an existing humanoid character controller (LMM (Holden et al., 2020)) with a CrossLoco-trained policy to facilitate interactive robot control without the need to retrain a large-scale interactive robot control policy from scratch. For more details, please refer to our supplementary videos and the appendix.

## 6 DISCUSSION

We introduce CrossLoco, an unsupervised reinforcement learning framework designed to enable robot locomotion control driven by human motion. This framework incorporates a cycle-consistency-based reward function, which facilitates the discovery of robot skills and the establishment of correspondence between human and robot motion. Our experimental results demonstrate the effectiveness of our approach in transferring a wide range of human motions to control a robot that has a different morphology.

Our next steps involve exploring two directions. Firstly, we aim to extend our framework beyond locomotion control to more complex scenarios, including long-horizon human demonstrations that involve long-distance locomotion and tool manipulation on a legged-manipulation robot, such as a quadrupedal robot with an arm mounted on its body. Secondly, we are interested in implementing our method on real-world robots for practical applications.

## ACKNOWLEDGMENTS

This work is supported by National Science Foundation under Grant No.2222723. We would like to thank Yuxiang Yang and Jiahan Fan for valuable suggestions.

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

## A  TRAINING HYPERPARAMETERS

In this study, we trained the policy using Proximal Policy Optimization (PPO) Schulman et al. (2017). Additionally, both the H2R-Mapper and R2H-Mapper were trained utilizing supervised learning techniques. Below, we detail the hyperparameters employed in our training process.

| | |
|---|---|
| Env number | 1024 |
| Batch size | 2048 |
| Policy lr | 1e-4 |
| Critic lr | 1e-4 |
| Mappers lr | 1e-4 |
| Optimizer | Adam |
| Clip | 0.2 |
| Entropy Coefficient | 5e-3 |
| $\gamma$ | 0.99 |
| $\lambda$ | 0.9 |

Table 2: Training hyperparameters.

## B  APPLICATIONS

Many research and work have explored the field of human motion synthesis using various input sources, including text (Bahl et al., 2022), music Tseng et al. (2023), and user inputs (Holden et al., 2020). Our learned human2robot controller can be seamlessly integrated with these modules by using human motion as the interface for new applications. In this section, we present two examples: **Language2Robot** and **Interactive Robot Control**.

**Language2Robot.** Our approach involves utilizing the text2human module in combination with our human2robot controller. This allows for the generation of robot movements from language by first producing human motion using the text2human module, which is then transferred to the robot using the human2robot controller. Our method differs from recent language to quadrupedal robot motion work (Tang et al., 2023) in that it does not require an engineering-intensive interface, such as foot contact patterns, which could limit the range of possible generated robot motions.

In our implementation, we utilize the Human Motion Diffusion Model (MDM) (Bahl et al., 2022) as our text-to-human motion translator. MDM is a diffusion model-based lightweight model that achieves state-of-the-art results on leading benchmarks. However, MDM uses AMASS (Mahmood

et al., 2019) human model which is different from the LaFAN1 (Harvey et al., 2020) model we used for training the policy. Therefore, we retarget the outputs of the MDM to the skeleton model we use.

Our study involves testing the generation of robot motion based on different input messages. The results of our study are presented in Figure 5. We show that this framework can generate robot motion according to instructions. For instance, the message "strides swiftly in a straight" results in a fast walking straight robot motion, while "squats down then jumps" triggers a robot squats and jumps motion.

**Interactive Robot Control.** Interactive control of robots in response to changing conditions or unexpected obstacles is a significant challenge in robotics, which involves careful controller design and motion planning. Recent advances in character animation enable users to interactively control human characters using joysticks, automatically adapting their motion styles to the surrounding environment, like crouching in confined spaces or leaping over obstacles. Our key idea for the second application, **Interactive Robot Control**, is to leverage the existing human animation techniques for robot control. Instead of retraining a large-scale model from scratch, we simply translate the output of the existing character controller to the robot's operational space using the proposed method.

Our implementation of this framework utilizes Learned Motion Matching (LMM) (Holden et al., 2020), which is a scalable neural network-based framework for interactive motion synthesis. We combine LMM with our learned controller. During interactive robot control, LMM takes input user commands for human motion generation, and our controller converts the generated human motion to robot control commands.

We evaluated our implementation by controlling the robot using a joystick. Figure 5 presents the results of using LMM2robot. The experiment demonstrated that the robot can actively adjust its motion based on the user's commands. These results provide evidence of the effectiveness of our learned controller for interactive robot control.

