# OpenReview forum: "CrossLoco: Human Motion Driven Control of Legged Robots via Guided Unsupervised Reinforcement Learning"
_ICLR.cc/2024/Conference — ICLR 2024 poster_

### Official Review · Reviewer_Xm7A · 2023-10-29

**Soundness:** 3 good
**Presentation:** 3 good
**Contribution:** 2 fair
**Rating:** 6
**Confidence:** 4

**Summary:**

This paper proposes learning cross-morphology human motion-driven control to map from human motion to robot control signals in a guided unsupervised learning fashion. Without using any pre-collected dataset from the robot domain, this work proposes using a cycle-consistency reward based on translation from human motion to robot motion and then from robot motion to human motion to train such a control network. By formulating the problem as a skill-discovery problem, where the skill is specified through direct human motion input and the reward is based on the mutual information between the robot state and human motion (specified by the cycle-consistency reward), the method shows interesting results in translating human motion to semantically meaningful and natural robot motion.

**Strengths:**

- This paper provides a convincing approach for automatic human motion to robot motion translation framework that does not rely on any predefined robot motion dataset. Since human motion is relatively accessible, using human motion to directly control robot motion and try to create semantically similar motion is an intuitive and useful approach.
- I find the use of cycle consistency to translate between human motion and robot motion a well-principled approach for this task. As far as I know, this is a novel approach. The R2H-Mapper and H2R-Mapper provide a scalable approach for human motion-guided robot control.
- The provided qualitative result shows that the method can translate human motion into relatively realistic human motion. In the handful of motion with semantic meaning (dancing motion), the robot generates similarly semantically meaningful motion.
- Compared to prior methods that perform retargeting in the kinematic space (e.g. ACE [1]), this paper tackles the more challenging simulated control task.

[1] Li, Tianyu et al. “ACE: Adversarial Correspondence Embedding for Cross Morphology Motion Retargeting from Human to Nonhuman Characters.” ArXiv abs/2305.14792 (2023): n. pag.

**Weaknesses:**

- Missing evaluation on the R2H-Mapper and H2R-Mapper. As the main reward provider, it is important to visualize how well these learned networks can translate between the two different modalities of motion. Do certain correspondences emerge (e.g. robot forward foot mapping to hands)? Are there failure modes that the mapper didn't learn?
    - I think the whole application section could be replaced with a more detailed analysis and results from the two mappers. If the method can really handle human motion, applying it to language control and interactive control could be explored to supplement or just provide video results. If applications are studied, I think the robustness of the method to different modes of human motion should be investigated: e.g. can the learned motion mapper handle unseen motion like breakdancing and cartwheeling?
    - Since the mapping between poses $p^h$ and $p^r$ is done locally, I think showing the reconstructed result together with the ground truth root would be beneficial.
- Though the qualitative results provide a number of motion sequences, most of them are locomotion sequences (walking and running) which I think a Task-only controller should solve relatively easily. The only semantically meaningful sequences are dancing, of which there are only a handful of examples. There is a glimpse of good semantic mapping, but it is not super conclusive. Instead of focusing on dancing, maybe simpler motions like raising hands, reaching for objects, and more complex hand movements could be more suitable for evaluating this system's capabilities. Similar to the previous point about evaluating the mappers, it's not clear how well each part of the human body is mapped.
    - The results on Deepmimic show that an expert-designed mapping may not be plausible and not easy to learn. However, since there is no study on what kind of mapping CrossLoco learned, it is hard to see how CrossLoco is better. Does a good mapping exist between bipedal and quadruped robots? Is this task too ill-posed? Or have we just not found a good mapping yet?
- Need to discuss the limitations of the current framework. On what human motion does the matching score the lowest? Which human motion can be best matched? What happens if more or fewer human motion sequences are used for learning the mapping?
    - I think the entire application section could be replaced with a more detailed analysis and results from the two mappers. If the method can truly handle human motion, it could be applied to language control and interactive control, which could be supported by supplementary materials or video results. If applications are studied, I believe the robustness of the method to different modes of human motion should be examined. For example, can the learned motion mapper handle unseen motions such as breakdancing and cartwheeling?
    - Since the mapping between poses $p^h$ and $p^r$ is done locally, I believe showing the reconstructed result together with the ground truth (GT) root would be beneficial.

**Questions:**

- I think providing results on the two Mappers, either a reconstruction metric or visualizations, could really help in understanding the capability of the framework. Does the algorithm learn some novel mapping?
- Showing results on simple and semantically meaningful sequences, such as raising hands, moving hands, and hands and feet, could really demonstrate the capabilities of the method. Showing results on locomotion does not really add much value since root tracking reward already exists.

---

> ### Author Response · Authors · 2023-11-22
> **Regarding Reviewer's Comments**
>
> Dear Reviewer Xm7A:
>
> We are grateful for your valuable comments. In the subsequent sections, we address the concerns you raised and provide a summary of the changes made to the paper. For your convenience in reviewing these modifications, we have attached a document that highlights the differences
>
> Q: I think providing results on the two Mappers, either a reconstruction metric or visualizations, could really help in understanding the capability of the framework. Does the algorithm learn some novel mapping?
>
> A:  We add a paragraph in the result section 5.1 to analyze and visualize the result of the learned mapping. Here we summarize our analysis and findings:
> - The mappers can accurately reconstruct human and robot poses when the robot actively follows human movement.
> - In scenarios where human motion is relatively stationary, human arm movements can be mapped onto the robot's front legs.
> - When mapping high-speed human locomotion, the system tends to overlook human arm motion and focuses primarily on leg motion. In such cases, the mapper cannot fully reconstruct the human pose.
> - When the variety of human motions is limited, the mappers can overfit to certain poses, leading to suboptimal motion transfer results.
>
> Q: Showing results on simple and semantically meaningful sequences, such as raising hands, moving hands, and hands and feet, could really demonstrate the capabilities of the method. Showing results on locomotion does not really add much value since root tracking reward already exists.
>
> A:  In our revised manuscript, we have included examples to visualize the results of mapping, which also encompass hand motion mapping. However, we also would like to emphasize the importance of results pertaining to locomotion as well. Our findings demonstrate that varying human movements can result in distinct robot motions, even when these human movements share the same root trajectory. For instance, humans can walk with different footstep frequencies, and correspondingly, the robot motions exhibit variations in footstep frequencies. Such diverse footstep strategies are beneficial for expressing human emotions. Our user study shows that users can distinguish the source human motion from the generated robot motion even though these robot motions have very similar root trajectories. This result indicates our method preserves human locomotion style and therefore leads to different locomotion behaviors.
>
> Q: Though the qualitative results provide a number of motion sequences, most of them are locomotion sequences (walking and running) which I think a Task-only controller should solve relatively easily. The only semantically meaningful sequences are dancing, of which there are only a handful of examples. There is a glimpse of good semantic mapping, but it is not super conclusive. Instead of focusing on dancing, maybe simpler motions like raising hands, reaching for objects, and more complex hand movements could be more suitable for evaluating this system's capabilities.
>
> A: We appreciate your valuable suggestion. We included additional example in the paper Figure 3 and a supplemental video regarding hand motion. Regarding the diversity of locomotion sequences compared to a Task-only controller, please refer to the answer to the previous question for more details.
>
> Q: Similar to the previous point about evaluating the mappers, it's not clear how well each part of the human body is mapped.
> The results on Deepmimic show that an expert-designed mapping may not be plausible and not easy to learn. However, since there is no study on what kind of mapping CrossLoco learned, it is hard to see how CrossLoco is better. Does a good mapping exist between bipedal and quadruped robots? Is this task too ill-posed? Or have we just not found a good mapping yet?
>
> A: We contend that our mapping approach is superior because it captures subtle and detailed nuances, such as variations in dancing or gait styles at the same speed, which [Kim et al. 2022] and [Li et al. 2023] fail to capture. Moreover, our concurrent training methodology based on reinforcement learning (RL) enables us to discover physically plausible motions, a feature not consistently present in previous methods. Finally, manually designing an effective, universal retargeting strategy for all types of motions is challenging. In contrast, CrossLoco effectively processes the human motion dataset as a whole to develop a universal strategy.
>
> Q: Need to discuss the limitations of the current framework. On what human motion does the matching score the lowest? Which human motion can be best matched? What happens if more or fewer human motion sequences are used for learning the mapping?
>
> A: We discussed several failure cases in section 5.1 paragraph 3. These cases involve human backward walking and swift side steps. In the section, we also have added more analysis on the cases where less diverse human motion sequences are used.

---

> > ### Comment · Reviewer_Xm7A · 2023-11-23
> > **Response to Authors**
> >
> > I thank the authors for the detailed response and additional result.
> >
> > I appreciate the result on raising front leg, it is helpful in showing that it learns semantics. I still have some doubts about the reconstruction process through; why not show some quantitative and qualitative result of the reconstructed human motion after the mapping process (that is human -> robot -> human) to better show it does learn the mapping.

---

> > > ### Author Response · Authors · 2023-11-23
> > > **New Illustration**
> > >
> > > We appreciate the reviewer's comments and have taken them into consideration.
> > >
> > > We have uploaded a revised version of the paper to address the suggested changes.
> > >
> > > In particular, we have included the reconstruction error of each pose in Figure 3. These values indicate that when the reconstruction error is low, there is a strong correspondence between human and robot motion, allowing the robot to accurately represent the human's motion. On the other hand, when the error is high, the correspondence between the two domains is weaker, leading the robot to disregard the human's arm motion.

---

> ### Author Response · Authors · 2023-11-22
> **Summary of Modifications**
>
> Additionally, we have attached a document highlighting the differences for your convenience in reviewing the changes.
>
> Changes:
> - We have enhanced the results section by adding analysis and visualizations of the mapping results.
> - The Application section has been shortened, with more detailed content shifted to the appendix.
> - Training hyperparameters have been added to the appendix for comprehensive understanding.
> - Additional motion results have been included in the supplementary videos.
> - Ambiguous terms throughout the paper have been clarified for improved clarity and precision.
>
>
> Sincerely,
> Authors of CrossLoco

---

### Official Review · Reviewer_QSkw · 2023-11-01

**Soundness:** 3 good
**Presentation:** 2 fair
**Contribution:** 3 good
**Rating:** 6
**Confidence:** 4

**Summary:**

This paper presents CrossLoco, an unsupervised reinforcement learning framework designed to translate human motions into robot controls, addressing the challenge of establishing correspondence between humans and quadrupeds. This framework introduces a cycle-consistency-based reward term and maximize mutual information between human motions and robot states. CrossLoco outperforms baseline algorithms demonstrating its effectiveness. Additionally, the framework shows use in applications like language2robot and interactive robot control.

**Strengths:**

The implementation of both R2H-Mapper and H2R-Mapper to reconstruct human and robot motion is a novel approach that ensures cycle consistency.
The inclusion of a supplementary video provides a more intuitive showcase of the qualitative results.
The authors conducted a user study to demonstrate the effectiveness of their motion retargeting control results.
The potential applications for language-driven motion control and interactive motion control for quadrupeds are promising. Scaling up to larger human more datasets to enable quadrupeds to perform more tasks could be very promising.

**Weaknesses:**

The baseline used for retargeting seems weak. Why not employ stronger learning-based retargeting baselines combined with motion imitation algorithms?

The definition of the root tracking reward is unclear, particularly the definition of $s_{root}$ and $\bar s_{root}$. I assume $s_{root}$ refer to robot root states including global translation and orientation but what is $\bar s_{root}$? The paper claims this reward minimizes deviation between the normalized base trajectory of human and robot, is  $\bar s_{root}$ the root states of human? A clearer definition would be beneficial.

The evaluation metrics could be improved. For instance, introducing Frechet Inception Distance (FID), used in ACE, to measure human motions and robot motions.

Additionally, the use of the averaged correspondence reward is questionable; as the author mentioned, a human forward walking motion could be mapped into a robot’s lateral movements. This metric may not accurately reflect correspondence between motions and seems more like a reflection of the training reward (loss) rather than a robust evaluation metric. A suggestion would be for the author to use paired data to train the two mappers independently and apply these mappers for testing all methods or directly use existing SOTA motion retargeting models.

There is only one sequence for qualitative comparison with the baselines, I encourage authors to provide more and include the reference motion for deepmimic (retargeting results).

**Questions:**

It seems the authors didn't provide details about the human motion dataset, also lack RL training and supervised training hyperparameters. These details would be beneficial for researchers to follow.

---

> ### Author Response · Authors · 2023-11-22
> **Regarding Reviewer's Comments**
>
> Dear Reviewer QSkw:
>
> We appreciate your review. In the following sections, we will address the concerns you raised and then present a summary of the changes made to the paper. Additionally, we have attached a document highlighting the differences for your convenience in reviewing the changes.
>
>
> Q: It seems the authors didn't provide details about the human motion dataset, also lack RL training and supervised training hyperparameters. These details would be beneficial for researchers to follow.
>
> A: We apologize for not providing sufficient information about the dataset and training hyperparameters in the original version. The dataset information can be found in Section 5 paragraph 2. We have also added more training information to the Appendix.
>
>
> Q: ‘The baseline used for retargeting seems weak. Why not employ stronger learning-based retargeting baselines combined with motion imitation algorithms?
>
> A: Learning-based retargeting baselines, such as ACE, typically require motion datasets from both domains to establish correspondence. While human datasets are usually accessible, robot datasets containing expressive motion are often unavailable. This lack of data hinders our ability to perform retargeting and imitation. In contrast, CrossLoco employs unsupervised reinforcement learning (RL) to implicitly learn the correspondence between the human and the robot. We have clarified this in the second paragraph of the introduction.
>
>
> Q: The definition of the root tracking reward is unclear, particularly the definition of $s^{root}$ and $\bar{s}^{root}$. I assume$s^{root}$ refer to robot root states including global translation and orientation but what is $\bar{s}^{root}$? The paper claims this reward minimizes deviation between the normalized base trajectory of human and robot, is $s^{root}$   the root states of human? A clearer definition would be beneficial.
>
> A:  As humans and robots have different leg lengths, we normalize the root position and root height, relative to the leg lengths of both humans and robots. Therefore sroot  and  sroot  represents this normalized root state of the robot and human. Additional clarification on this has been added to the paper Section 4.3 paragraph 3.
>
> Q: The evaluation metrics could be improved. For instance, introducing Frechet Inception Distance (FID), used in ACE, to measure human motions and robot motions.
>
> A: We argue that FID is not a suitable evaluation metric in our setting. The FID metric evaluates the difference between two distributions. The ACE paper utilizes the FID score to compare distributions within a shared feature space (end-effector space). However, in CrossLoco, there is no predefined space that is shared between robots and humans. We considered measuring the FID between the distribution of reconstructed human poses and the distribution of input human motions. However, because the correspondence reward already includes measuring the distance between distributions, we believe it is unnecessary to add an FID score.
>
> Q: Additionally, the use of the averaged correspondence reward is questionable; as the author mentioned, a human forward walking motion could be mapped into a robot’s lateral movements. This metric may not accurately reflect correspondence between motions and seems more like a reflection of the training reward (loss) rather than a robust evaluation metric. A suggestion would be for the author to use paired data to train the two mappers independently and apply these mappers for testing all methods or directly use existing SOTA motion retargeting models.
>
> A: The average correspondence reward is assessed to determine whether the robot's motions correspond to human movements. In other words, it evaluates whether the robot moves in sync with human motion. A high correspondence reward signifies that different human motions lead to distinct robot movements. For instance, good correspondence should not map different human dancing poses into a single robot pose. While there could be multiple correspondence solutions, we have incorporated a root tracking reward as 'guidance' for constructing this correspondence.
> Regarding the suggestion to pre-train mappers with paired data before using them for training, we acknowledge it as a viable approach for a small dataset. However, the primary challenge with this method lies in the manual construction of paired data, particularly motion pairs that are physically feasible for the robot. We believe a key contribution of CrossLoco is its ability to build correspondence without relying on a robot dataset.
>
> Q: There is only one sequence for qualitative comparison with the baselines, I encourage authors to provide more and include the reference motion for deepmimic (retargeting results).
>
> A: We add 2 examples to our supplementary video. Please check out the new video 2:15-2:36.
>
>
> Please check out the new supplementary material for modification of paper and new video.
>
> Sincerely,
> Authors of CrossLoco

---

> ### Author Response · Authors · 2023-11-22
> **Summary of Modifications**
>
> Additionally, we have attached a document highlighting the differences for your convenience in reviewing the changes.
>
> Changes:
> - We have enhanced the results section by adding analysis and visualizations of the mapping results.
> - The Application section has been shortened, with more detailed content shifted to the appendix.
> - Training hyperparameters have been added to the appendix for comprehensive understanding.
> - Additional motion results have been included in the supplementary videos.
> - Ambiguous terms throughout the paper have been clarified for improved clarity and precision.

---

### Official Review · Reviewer_TXup · 2023-11-03

**Soundness:** 3 good
**Presentation:** 3 good
**Contribution:** 3 good
**Rating:** 8
**Confidence:** 3

**Summary:**

The paper proposes the CrossLoco framework, a guided unsupervised reinforcement learning framework that simultaneously learns robot skills and their correspondence to human motions. This is achieved thanks to the use of a "cycle-consistency-based" reward, inspired in generative vision system such as Cycle-GAN. The reward function has components related to the learning of a policy, components related to the human-robot correspondence problem, and terms to regulate the training and preserve the high-level semantics. The proposed framework is validated in the task of transferring a set of human motions to the Aliengo quadrupedal robot, and its performance compared against manually engineered and unsupervised base-line algorithms. CrossLoco obtains better results than the other algorithms in this comparison.

**Strengths:**

1. The main contribution of this paper is the CrossLoco framework which uses a "cycle-consistency-based" reward. The reward function has components related to the learning of a policy, components related to the human-robot correspondence, and additional rewards terms to regulate the training and preserve the high-level semantics.

2. The proposed framework is correctly validated by using it for transferring a set of human motions to the Aliengo quadrupedal robot.

3. The paper is clear, well-organized and well-written.

**Weaknesses:**

I do not see any clear weakness.

**Questions:**

No questions

**Details Of Ethics Concerns:**

No ethics concerns

---

> ### Author Response · Authors · 2023-11-22
> **Summary of Modifications**
>
> Dear Reviewer TXup:
>
> We appreciate your review. Here we present a summary of the changes made to the paper. Additionally, we have attached a document highlighting the differences for your convenience in reviewing the changes.
>
> Changes:
>
> - We have enhanced the results section by adding analysis and visualizations of the mapping results.
> - The Application section has been shortened, with more detailed content shifted to the appendix.
> - Training hyperparameters have been added to the appendix for comprehensive understanding.
> - Additional motion results have been included in the supplementary videos.
> - Ambiguous terms throughout the paper have been clarified for improved clarity and precision.
>
> Sincerely,
>
> Authors of CrossLoco

---

### Meta-Review · Area_Chair_e2pg · 2023-12-06

**Metareview:**

Summary: This paper studies the problem of finding correspondences between human bodies (and motion) and robot morphology (and motion). The key idea is a cycle-consistency-based reward term which maximizes the mutual information between human motions and robot states.The paper demonstrates how CrossLoco translates  human motions such as running, hopping, and dancing onto a quadrupedal robot.


Strengths: Reviewers agreed that the cycle-consistency reward is novel, and the results are compelling. After rebuttals, the paper also included helpful and careful ablations on how different isolated body part movements on the human translate to motion on the robot quadruped.

Weaknesses: It would be compelling to see more robot morphologies in addition to the quadruped (e.g., quadrotor, fixed-base manipulator) to push the “limits” of the approach and how well-posed the problem is of transferring motions between morphologies.

**Justification For Why Not Higher Score:**

While this work is interesting and well-executed, I didn't give a higher score because the impact of this work was only demonstrated on a single robot morphology; the impact would have been bigger with multiple.

**Justification For Why Not Lower Score:**

After I carefully reviewed the manuscript and the author-reviewer discussion, I agree with the reviewers that this is an interesting piece of work and recommend acceptance.

---

### Decision · Program_Chairs · 2024-01-16

Accept (poster)